# Risk Factors for Prolonged Mechanical Ventilation and Delayed Extubation Following Bimaxillary Orthognathic Surgery: A Single-Center Retrospective Cohort Study

**DOI:** 10.3390/jcm11133829

**Published:** 2022-07-01

**Authors:** Christian I. Schwer, Teresa Roth, Mathieu Gass, René Rothweiler, Torsten Loop, Marc C. Metzger, Johannes Kalbhenn

**Affiliations:** 1Department of Anesthesiology and Intensive Care Medicine, University Medical Center Freiburg, 79106 Freiburg, Germany; torsten.loop@uniklinik-freiburg.de (T.L.); johannes.kalbhenn@uniklinik-freiburg.de (J.K.); 2Department of Oral and Maxillofacial Surgery, University Medical Center Freiburg, 79106 Freiburg, Germany; teresa.roth1703@googlemail.com (T.R.); mathieu.gass@uniklinik-freiburg.de (M.G.); rene.rothweiler@uniklinik-freiburg.de (R.R.); marc.metzger@uniklinik-freiburg.de (M.C.M.)

**Keywords:** bimaxillary surgery, airway complications, risk factors, extubation

## Abstract

Background: Bimaxillary orthognathic surgery bears the risk of severe postoperative airway complications. There are no clear recommendations for immediate postoperative follow-up and monitoring. Objective: to identify potential risk factors for prolonged mechanical ventilation and delayed extubation in patients undergoing bimaxillary orthognathic surgery. Methods: The data of all consecutive patients undergoing bimaxillary surgery between May 2012 and October 2019 were analyzed in a single-center retrospective cohort study. The clinical data were evaluated regarding baseline characteristics and potential factors linked with delayed extubation. Results: A total of 195 patients were included; 54.9% were female, and the median age was 23 years (IQR 5). The median body mass index was 23.1 (IQR 8). Nine patients (4.6%) were of American Society of Anesthesiologists Physical Status Classification System III or higher. The median duration of mechanical ventilation in the intensive care unit was 280 min (IQR, 526 min). Multivariable analysis revealed that premedication with benzodiazepines (odds ratio (OR) 2.60, 95% confidence interval (0.99; 6.81)), the male sex (OR 2.43, 95% confidence interval (1.10; 5.36)), and the duration of surgery (OR 1.54, 95% confidence interval (1.07; 2.23)) were associated with prolonged mechanical ventilation. By contrast, total intravenous anesthesia was associated with shorter ventilation time (OR 0.19, 95% confidence interval (0.09; 0.43)). Conclusion: premedication with benzodiazepines, the male sex, and the duration of surgery might be considered to be independent risk factors for delayed extubation in patients undergoing bimaxillary surgery.

## 1. Introduction

Bimaxillary orthognathic surgery is performed to correct significant dental malocclusion, and to restore esthetic facial contour and proportion [1]. It reduces temporomandibular joint symptoms and plays a pivotal role in the treatment of obstructive apnea [2,3].

Considering the literature, bimaxillary surgery is a safe and reliable procedure, and the rate of intra- and postoperative complications is rather low [4,5]. However, Kantar et al. have recently reported that, compared to single-jaw surgery, double-jaw osteotomies are associated with an increased risk of early complications and surgery in the outpatient setting, and patients of American Society of Anesthesiology (ASA) physical status class 3 or higher have been identified as independent factors for postoperative adverse effects [6]. Taking this into account, there are no clear guidelines or recommendations for immediate postoperative follow-up and monitoring in patients undergoing bimaxillary surgery. With the goal of avoiding early severe postoperative complications owing to nasal airway obstruction, edema, or intraoral bleeding, delayed controlled extubation in the ICU may be an approach after bimaxillary surgery. However, prolonged nasotracheal intubation bears the risk of adverse effects such as epistaxis, turbinectomy, retropharyngeal dissection, tympanites, and nasal alar pressure ulcers [7], and prolonged mechanical ventilation was linked with transient dysphonia, dysphagia, sore throat, and pneumonia [8,9].

While many studies have addressed the issue of predictors for postoperative wound complications [10,11,12], risk factors for delayed extubation in patients undergoing bimaxillary surgery are poorly defined to date. Therefore, the purpose of this study was to evaluate the duration of mechanical ventilation and to identify potential risk factors for delayed extubation.

## 2. Materials and Methods

### 2.1. Study Design, Setting, and Participants

This retrospective cohort study was conducted at the Department of Anesthesiology and Intensive Care, and the Department of Oral and Maxillofacial Surgery and Regional Plastic Surgery, University Medical Center, Freiburg, Germany. The study protocol was approved by the Ethics Committee of Freiburg University Medical Center (approval number 200/20). This article adheres to the Strengthening the Reporting of Observational Studies in Epidemiology (STROBE) guidelines [13]. A STROBE checklist has been provided in the Appendix A. The study was initiated in 2020, and the retrospective data collection was conducted in 2020. Due to the initiation of an electronic patient data and management system in 2012 that allows for gaining the relevant data, we enclosed only files from 2012 or later. The study cohort consisted of all patients who had undergone bimaxillary orthognathic surgery followed by admission to the ICU between May 2012 and October 2019. The observational retrospective study design removed the need for a priori sample size calculation.

### 2.2. Anesthesia, Postoperative Care, and ICU Therapy

Patients fasted for 6 h for solid food and 2 h for clear liquid prior to the planned induction of anesthesia. If desired, patients received 3.75 or 7.5 mg midazolam orally before being transferred to the operating theater. Anesthesia was induced with the i.v. application of remifentanil, propofol, and cisatracurium, and maintained with propofol or volatile anesthetic sevoflurane or desflurane. Noninvasive arterial blood pressure, electrocardiography, and pulse oximetry were monitored continuously. Gastric feeding tube placement was performed in all patients. In order to control and reduce postoperative swelling after orthognathic surgery [14], patients who had no contraindications received a single preoperative high-dose injection of dexamethasone.

All patients were transferred to the intensive care unit (ICU) in a sedated state with a continuous i.v. application of propofol (doses in the range of 80–120 µg/kg/min) under controlled mechanical ventilation and intubated endotracheally for planned extubation. Local cooling of the midface and the neck with ice packs or an automated cooling mask (Hilotherapy^®^, Hilotherm GmbH, Argenbühl-Eisenharz, Germany) was consequently applied. Sedation was stopped with stable vital parameters, decayed muscle relaxant, and analgesic therapy with nonsteroidal drugs, and nurse-controlled opioid application was established. Desired sedation depth was between –1 and 0 using the Richmond agitation and sedation scale [15]. When patients were alert and calm, the standard operating procedure for the extubation of patients undergoing bimaxillary orthognathic surgery was applied. The main premises are the evaluation of a patient’s ability to cough, swallow, and cooperate, and a successful leakage test with a deflated cuff of the endotracheal tube (Figure 1).

### 2.3. Surgical Protocol

Each patient preoperatively received orthodontic treatment. The bimaxillary surgeries were performed under general anesthesia with nasal intubation. The virtual planning of the surgery was performed using Dolphin software (Patterson Dental, Chatsworth, CA, USA), and the surgical splints were printed out with a Stratasys Eden 260v 3D printer (Stratasys, Eden Prairie, MN, USA).

After applying local anesthesia with adrenaline 1:200,000 in the maxilla and the mandible, the surgery started in the maxilla with a leFort-I osteotomy. After repositioning, the maxilla was fixed with 4 L plates 1.5 mm and 16 Cortical Screws 2.0/6 mm (DePuy Synthes, Raynham, MA, USA); in the case of a gap, BioOss Kollagen (GeistlichPharma, Wolhusen, Switzerland) was added to the osteotomy line. In the mandible, bilateral sagittal split osteotomy (BSSO) was performed following the Obwegeser/Dal Pont technique; after adjustment, the newly positioned mandible was fixed with two SplitFix 2/40 mm plates and 4 cortical screws 2.0/6 mm (DePuy Synthes, Raynham, MA, USA). In cases with large mandibular advancements, an additional osteosynthesis plate was used in the mandible to increase overall stability in comparison to SplitFixPlate alone. In the case of a maxillary and mandibular advancement because of sleep apnea or when performing counterclockwise rotation, the surgery was performed following a mandible first protocol.

### 2.4. Data Collection

To determine factors associated with the extubation period, the case records were reviewed for general demographic data, and specific medical, operative, and anesthesia predictor variables. Inclusion criteria for this study were patients with a developmental dentofacial deformity involving the two jaws. Demographic variables were age at the time of operation and gender. The medical variables were pre-existing comorbidities, ASA classification, Mallampati score, and body mass index (BMI). Operative and anesthesia variables were surgery duration and types of drugs used. As the primary outcome, variable time to extubation on ICU was defined.

### 2.5. Data Analyses

The data were collected in a MS Excel™ (Microsoft, Redmond, WA, USA) datasheet. Further statistical processing was performed using the Statistical Package for the Social Sciences (SPSS for Windows, V.27; SPSS Inc., Chicago, IL, USA).

Descriptive statistics were used to show the distribution of variables (median and range for continuous variables, and frequency for discrete variables). The quartiles of postoperative mechanical ventilation intervals were calculated. After that, two groups were formed: The “short-term postoperative mechanical ventilation interval” group, comprising the lower three quartiles, and the “long-term postoperative mechanical ventilation interval” group, comprising all patients with ventilation times longer than the 75th percentile. Normal distribution was tested using the Kolmogorov–Smirnov test. For the comparison of metric parameters between the two groups, such as duration of surgery, volume intake, and blood loss, a t-test for independent samples was used; for the comparison of nominal parameters such as sex and comorbidities, a chi-squared test was applied. A *p*-value of 0.05 was chosen to be the level of significance. To find the variables independently associated with longer postoperative ventilation, parameters with significant differences were included in binary logistic regression analysis. If the *p* value was less than 0.05, it was considered to be significant. For variable selection, the forward stepwise approach was applied.

## 3. Results

### 3.1. Preoperative Variables

The patients’ characteristics are shown in Table 1. A total of 195 consecutive patients who had undergone bimaxillary surgery between May 2012 and October 2019 were included in this retrospective study. The patients’ median age was 23 years (IQR 8; range from 18 to 61 years), and 107 patients (54.9%) were female. The median BMI was 23.1 (IQR 5.0). Nine patients (4.6%) were of ASA class 3 or higher. Fifteen patients (7.7%) had a Mallampati score of III or higher. Potentially relevant comorbidities included hypertension (3.1%), allergic asthma (12.3%), chronic obstructive pulmonary disease (COPD (1.0%)), hypothyroidism (4.6%), depression (5.1%), or a history of smoking (13.8%).

### 3.2. Anesthesia and Operative Variables

Of the patients, 125 (64.1%) had received premedication in the form of oral midazolam before they were transferred to the operating theater (Table 2). Airway management during surgery was successfully accomplished with nasotracheal intubation in all cases. In total, 179 patients (91.8%) received a single injection of dexamethasone. The maintenance of general anesthesia using propofol-based total intravenous anesthesia (TIVA) was performed in 135 patients (69.2%). The median open-wound operating time was 238 min (IQR, 95 min). The median time of mechanical ventilation in the operating theater from the start of anesthesia till ICU arrival was 330 min (IQR, 106 min).

### 3.3. Time of Mechanical Ventilation in the ICU

As shown in Figure 2, the median time of mechanical ventilation in ICU was 280 min (IQR, 526 min).

Consequently, the need for endotracheal intubation for more than 665 min (75th percentile) was defined as prolonged mechanical ventilation. Of the patients, 48 (32.7%) underwent delayed extubation.

### 3.4. Statistical Analysis of Risk Factors and Outcome Variables

Next, we statistically analyzed potential risk factors for prolonged mechanical ventilation in the ICU (Figure 3).

Multivariable analysis revealed that the factor most strongly associated with delayed extubation in the ICU was premedication with benzodiazepines (odds ratio (OR) 2.60, 95% confidence interval (0.99; 6.81)), followed by the male sex (OR 2.43, 95% confidence interval (1.10; 5.36)), and the duration of surgery (OR 1.54, 95% confidence interval (1.07; 2.23)), whereas the maintenance of general anesthesia with propofol-based TIVA was associated with earlier extubation (OR 0.19, 95% confidence interval (0.09; 0.43)).

## 4. Discussion

Orthognathic surgery is a common and mostly safe procedure for correcting dentofacial deformities and malocclusions [17]. Risks of surgery include relapse of the jaw, jaw fracture, nerve injury, wound infection, or excessive blood loss, and the patient´s airway may be threatened by obstruction, edema, or intraoral bleeding [18,19,20]. Kantar et al. have recently shown that, compared with single-jaw surgery, double-jaw osteotomies are associated with significantly higher rates of overall complications. In this study, surgery in the outpatient setting and patient ASA physical status class 3 or higher were identified as independent risk factors for postoperative adverse effects in patients undergoing bimaxillary surgery [6]. Most complications occur early after the operation, and delayed extubation in the ICU has become the standard approach in our institution. While early extubation after bimaxillary surgery is a safe procedure and is associated with reduced ICU length and hospital stay [21], risk factors for prolonged mechanical ventilation and delayed extubation in this patient cohort are poorly defined to date.

In our study, anxiolytic premedication with oral midazolam was associated with prolonged mechanical ventilation and delayed extubation in the ICU despite short elimination half-life midazolam reducing psychomotor performance in healthy volunteers for several hours [22]. Interestingly, until now, there was only low-quality evidence that midazolam reduces anxiety when administered as the sole sedative agent prior to a medical procedure [23]. In geriatric patients undergoing brief surgical procedures, midazolam administration significantly prolonged postanesthesia care unit discharge time [24]. Mohammadi et al. have recently shown that oral premedication with clonidine might have beneficial effects in patients undergoing bimaxillary surgery [25]; in hypertensive patients, dexmedetomidine premedication provides better hemodynamic stability than that of midazolam [26].

In our study, the male sex was associated with delayed extubation in the ICU. The reason for this observation remains unclear. However, there is growing evidence of sex-specific differences in mechanically ventilated patients [27,28,29], and a retrospective study on hospitalized patients in an ICU showed that women had significantly shorter duration of mechanical ventilation, time to withdrawal of sedation, and time to onset of active exercises [30].

Another result of our study was that, compared to balanced anesthesia with volatile anesthetics, the maintenance of general anesthesia with propofol-based TIVA was associated with a shorter period of mechanical ventilation in the ICU. All patients included in our study were transferred to the ICU with continuous propofol i.v. application. A possible explanation for the observed difference may be that patients undergoing anesthesia with volatile anesthetics may need higher doses of propofol for the transfer to the ICU. Whether causal or not, in the TIVA-group, propofol infusion was continued for transport with a lower target concentration. Other beneficial and advantageous effects of TIVA over inhalational agents in the perioperative setting include reduced PONV and better analgesia, both resulting in greater patient satisfaction and shortened intubation time [31]. Thus, anesthesia maintenance with propofol might be advantageous in patients undergoing bimaxillary surgery.

As one would expect, the type of orthognathic surgery and the amount of mandibular advancement or setback may influence the postoperative mechanical ventilation time. Riekert et al. have recently shown that an early extubation strategy was associated with a shortening of ICU and inhospital stay, whereas postoperative complications such as nausea and vomiting, anemia, or respiratory dysfunction were not increased compared to a delayed extubation strategy in patients undergoing bimaxillary surgery [21]. Another result of this study was that the reduction in pharyngeal airway space did not increase the complication rate in this patient cohort. In our study, the duration of surgery correlated with intubation time in the ICU. This result is consistent with those of previous studies in which prolonged surgery was associated with delayed extubation [32,33]. Surgical procedures represent a potential trigger for systemic inflammation [34], and prolonged surgery increases the secretion of proinflammatory cytokines and endothelial dysfunction. As a result, postoperative swelling and airway obstruction may occur [35]. Due to a previously published protocol-based evaluation for the feasibility of extubation [16], none of the 195 patients included in our study required reintubation in the ICU.

The potential strengths of the study are that the study cohort consisted of all patients who had undergone bimaxillary orthognathic surgery followed by admission to the ICU between May 2012 and October 2019, and that all operations were performed by a single surgical team. However, our study has several limitations. First, the single-center design with the small sample of patients might limit the generalizability of the results. Second, due to its retrospective character, there might be an absence of data on potential confounding factors. Lastly, our findings are merely an association and cannot imply causation. Further randomized trials should be undertaken to assess predictors for delayed extubation, helping us in identifying patients more likely to undergo prolonged mechanical ventilation.

In conclusion, this study showed that premedication with midazolam, the male sex, and the duration of surgery are associated with prolonged mechanical ventilation and delayed extubation in ICU, whereas the maintenance of general anesthesia with propofol-based TIVA is associated with earlier extubation in patients undergoing bimaxillary orthognathic surgery.

## Figures and Tables

**Figure 1 jcm-11-03829-f001:**
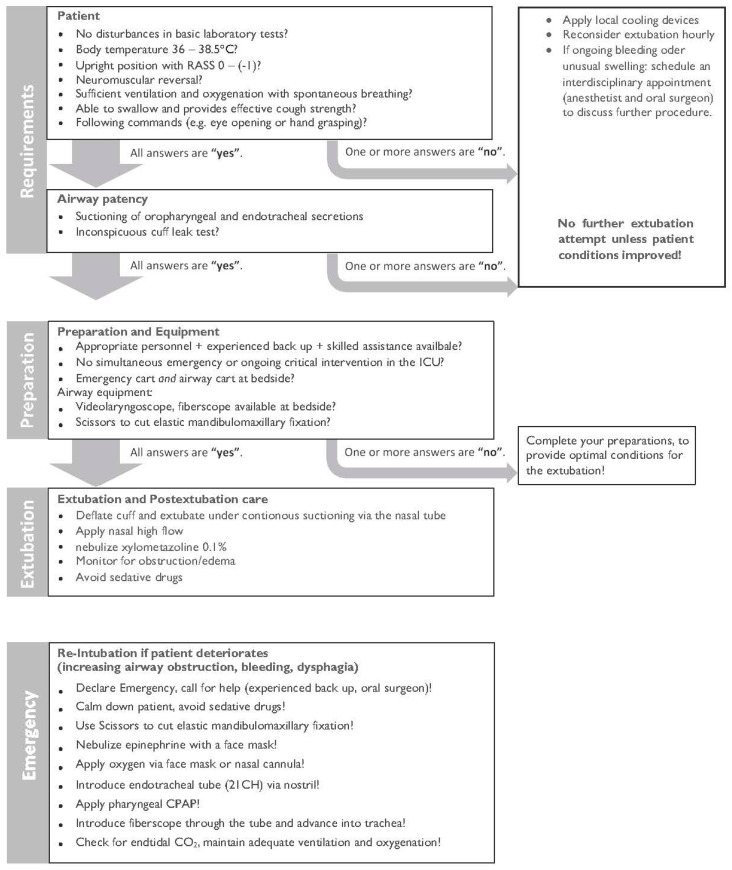
Standard operating procedure for the extubation of patients undergoing bimaxillary orthognathic surgery. Modified from standard operating procedure extubation of a difficult airway published in [16] by Schmutz et al.

**Figure 2 jcm-11-03829-f002:**
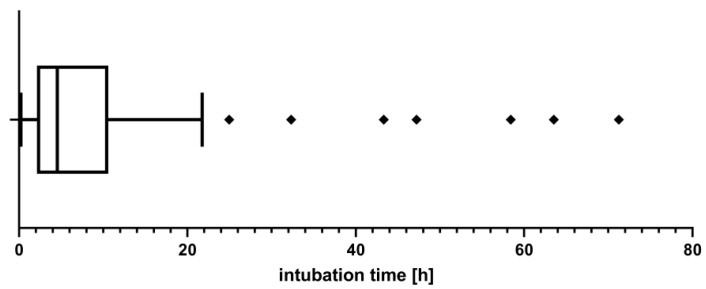
Box plot showing intubation time in ICU.

**Figure 3 jcm-11-03829-f003:**
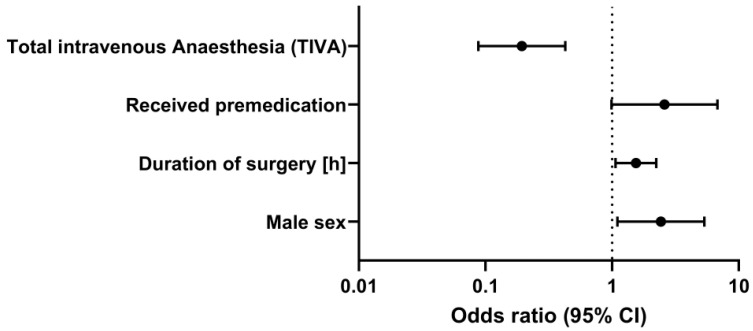
Risk factors for prolonged mechanical ventilation in the ICU.

**Table 1 jcm-11-03829-t001:** Patient characteristics.

Entire Cohort	*n* = 195
Age in years, median (IQR)	23.0 (8)
BMI, median (IQR)	23.1 (5)
Gender	
Male, *n* (%)	88 (45.1)
Female, *n* (%)	107 (54.9)
ASA classification, *n* (%)	
- I and II	186 (95.4)
- III–V	9 (4.6)
Mallampati grading, *n* (%)	
- 1 and 2	151 (77.4)
- 3 or higher	15 (7.7)
- Mallampati missing	29 (14.9)
Preexisting comorbidities, *n* (%)	
- Hypertension	6 (3.1)
- Allergic asthma	24 (12.3)
- COPD	2 (1.0)
- Hypothyroidism	9 (4.6)
- Depression	10 (5.1)
- Smoker	27 (13.8)

Categorical variables are given as absolute number and percentage. Continuous variables are given as median (IQR (interquartile range)). *ASA*, American Society of Anesthesiologists; BMI, body mass index (kg/m^2^); *COPD*, chronic obstructive pulmonary disease.

**Table 2 jcm-11-03829-t002:** Perioperative variables.

Entire Cohort	*n* = 195
Received premedication	125 (64.1%)
Intraoperative comedication, *n* (%)	
- Parecoxib	30 (15.4)
- Metamizole	15 (7.7)
- Tranexamic acid	19 (9.7)
Preoperative dexamethasone, *n* (%)	
- None	16 (8.2)
- 4 mg	10 (5.1)
- 8 mg	4 (2.1)
- 16 mg	20 (10.3)
- 20 mg	83 (42.6)
- 40 mg	49 (25.1)
- 44 mg	6 (3.1)
- 80 mg	6 (3.1)
- 84 mg	1 (0.5)
Intraoperative blood loss (mL), median (IQR)	300 (280)
Intraoperative fluid intake (mL), median (IQR)	1700 (1550)
Anesthesia maintenance, *n* (%)	
- Balanced anesthesia	60 (30.8)
- Total intravenous anesthesia	135 (69.2)
Time intervals (min), median (IQR)	
- Contact anesthesia until the start of surgical preparation	30 (15)
- Length of operation	238 (95)
- Mechanical ventilation until ICU arrival	330 (106)

Categorical variables are given as absolute number and percentage. Continuous variables are given as median (IQR (interquartile range)). *ICU,* Intensive Care Unit.

## Data Availability

Not applicable.

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
