# Peer review of "Risk Factors for Prolonged Mechanical Ventilation and Delayed Extubation Following Bimaxillary Orthognathic Surgery: A Single-Center Retrospective Cohort Study"

_jcm, 2022, doi:10.3390/jcm11133829_

Round 1

Reviewer 1 Report

Schwer et al present a study regarding predictors of postoperative delayed extubation in orthognatic surgery. The study is well designed and performed, and it addresses an interesting and unsolved topic: the need or not of postoperative mechanical ventilation in orthognatic surgery.

Although the manuscript is well performed, I have the following concerns, in order to improve the manuscript:

-Abstract: it should be structured. Include if possible OR and 95% CI of independent risk factors of delayed extubation

-Material and methods:

      -Please include number of approval of review board.

      -Reference the protocol in figure 1 (already published)

      -"All patients were transferred to ICU in a sedated state with a continuous iv application of propofol" According to results and discussion this sentence might not be true, please clarify what sedation was applied to patients that underwent inhalation anesthesia before ICU. Were IV benzodiazepines used?

   -Data analyses: "...were included in a binary logistic regression analysis to exactly calculate the risk for long term postoperative ventilation". It would be better: "to find the variables independently associated with longer postoperative mechanical ventilation"

-Results:

     -How many patients uderwent delayed extubation?

     -Figure 2: please clarify in the foot figure the box plot.

     -Multivariate analysis: what variables were introduced in the model?

-Discussion:

A study published in 2019 concluded that early extubation in the OR was safe and reduced lenght of stay. Pleased comment on that finding and discuss on the need or not of postoperative mechanical ventilation in this kind of surgery. Riekert M, Kreppel M, Schier R et al. Zöller JE, Rempel V, Schick VC. Postoperative complications after bimaxillary orthognathic surgery: A retrospective study with focus on postoperative ventilation strategies and posterior airway space (PAS). J Craniomaxillofac Surg. 2019 Dec;47(12):1848-1854. doi: 10.1016/j.jcms.2019.11.007.

Comment on the employed sedation in the patients who underwent inhalation anesthesia before transferring them to ICU.

Thank you for your valuable work

Reviewer 2 Report

Dear authors,

I think that the paper is interesting.

I wonder if the type of orthognathic surgery (class II, class III malocclusion) and the amount of mandibular advancement/setback, the number of  may influence the post-operative mechanical ventilation, as well as the number of screws.
There is a sentence saying "Depending on the case, an additional osthesynthesis plate was used in the mandible for increased stability".Even if there is no evidence regardig the type of intervention and the duration of post-operative mechanical ventilation, the study sample should be as much homogeneous as possible.
TITLE: I would suggest to add the study design.
DISCUSSION: I would suggest to add a section with "strenghts and limitations" of the study.
GUIDELINES: I would suggest to rport the study according to the STROBE guidelines, to fill the checklist and add it as supplementary material.
